

# Identifying the source rookery of green turtles (*Chelonia mydas*) found in feeding grounds around the Korean Peninsula

Min-Woo Park[1],[*], Il-Hun Kim[2],[*], Jaejin Park[1], Changho Yi[2], Min-Seop Kim[2], In-Young Cho[2], Il-Kook Park[1],[2], Hee-Jin Noh[3],[4], Sang Hee Hong[3],[4] and Daesik Park[1]

[1] Science Education, Kangwon National University, Chuncheon, Kangwon-do, Republic of South Korea
[2] Ecology and Conservation, National Marine Biodiversity Institute of Korea, Seocheon, Chungcheongnam-do, Republic of South Korea
[3] Ecological Risk Research Department, Korea Institute of Ocean Science and Technology, Geoje, Gyeongsangnam-do, Republic of South Korea
[4] Department of Ocean Science, University of Science and Technology, Daejeon, Republic of South Korea
[*] These authors contributed equally to this work.

Corresponding author
Daesik Park, parkda@kangwon.ac.kr

## ABSTRACT

Determining the genetic diversity and source rookeries of sea turtles collected from feeding grounds can facilitate effective conservation initiatives. To ascertain the genetic composition and source rookery, we examined a partial sequence of the mitochondrial control region (CR, 796 bp) of 40 green turtles (*Chelonia mydas*) collected from feeding grounds around the Korean Peninsula between 2014 and 2022. We conducted genetic and mixed-stock analyses (MSA) and identified 10 CR haplotypes previously reported in Japanese populations. In the haplotype network, six, three, and one haplotype(s) grouped with the Japan, Indo-Pacific, and Central South Pacific clades, respectively. The primary rookeries of the green turtles were two distantly remote sites, Ogasawara (OGA) and Central Ryukyu Island (CRI), approximately 1,300 km apart from each other. Comparing three parameters (season, maturity, and specific feeding ground), we noted that OGA was mainly associated with summer and the Jeju Sea, whereas CRI was with fall and the East (Japan) Sea ground. The maturity did not show a distinct pattern. Our results indicate that green turtles in the feeding grounds around the Korean Peninsula originate mainly from the Japan MU and have genetic origins in the Japan, Indo-Pacific, and Central South Pacific clades. Our results provide crucial insights into rookeries and MUs, which are the focus of conservation efforts of the Republic of Korea and potential parties to collaborate for green turtle conservation.

## INTRODUCTION

Management of sea turtle breeding and feeding grounds under the regional management unit framework is an effective approach to conserving endangered sea turtles (*Wallace et al., 2010*). The management unit (MU) exhibits distinct demographic processes, such as genetic composition and life history, and is a functionally independent unit for turtle conservation (*Mortiz, 1994*). Various conservation efforts have been conducted in breeding populations, such as protecting breeding sites, releasing captive-breeding turtles, and running educational programs (*Hamann et al., 2010*; *Barbanti et al., 2019*). Considering the frequent site fidelity of sea turtles for breeding and feeding grounds over their lifespan, the protection and management of both these sites are crucial for sea turtle conservation (*Hamann et al., 2010*). Sea turtles from different rookeries often gather on feeding grounds, making it important to protect multiple rookeries (*Nishizawa et al., 2013*; *Piovano et al., 2019*). Therefore, conservation efforts for feeding grounds need to be increased. Understanding the interaction between breeding and feeding grounds can be difficult due to sea turtles' wide distribution and complex life cycle. However, recent advancements in genetic analyses have enabled researchers to gain information about the turtles' genetic variability, composition, and origin. To assess these, both genetic analyses and mixed stock analysis (MSA) have been conducted on turtles captured or incidentally collected (*e.g.*, bycatch or stranded) (*Nishizawa et al., 2013*; *Piovano et al., 2019*). Previous studies on feeding grounds were mainly conducted in easily accessible or abundant areas where breeding grounds were nearby (*Shamblin et al., 2012*; *Read et al., 2015*). Among the seven known sea turtles, green turtles (*Chelonia mydas*) are the most common and are found across tropical and subtropical oceans worldwide (*Seminoff et al., 2015*). The species is listed as an endangered species in the IUCN Red List of Threatened Species (*IUCN, 2023*) and is highly migratory. A recent study (*Jensen et al., 2019*) identified 11 phylogenetic clades of green turtles worldwide; it showed that clade VIII, which includes most turtles in rookeries in the Indo-Pacific and South Western Indian Ocean MU, has the widest distribution range in the Pacific and Indian Oceans. In the Western Pacific Ocean, six clades (III, IV, V, VI, VII, and VIII) were defined, and clades VII, VIII, III, V, and VI have main haplotypes for Japan, Indo-Pacific (IP), Central Western Pacific (CWP), South Western Pacific (SWP), and Central South Pacific (CSP) MU, respectively (*Jensen et al., 2019*). The regional boundaries of these genetic clades aligned well with the existing MU of green turtle rookeries (*Wallace et al., 2010*), probably because of their high level of fidelity to breeding and feeding grounds (*Nishizawa et al., 2011*). Some turtles in the Japan MU also had genetic components belonging to clades III and VIII (*Hamabata, Kamezaki & Hikida, 2014*), which were found in the IP and CWP MU, while those in the IP MU also had components of clades III and VII. To understand the demographic structure of sea turtle populations in breeding and feeding grounds, and their connectivity with nearby MUs, studies on their phylogenetic origins are also necessary.

In the northwestern Pacific, several green turtle rookeries exist, namely, Ogasawara (OGA), Central Ryukyu Island (CRI), and Yaeyama Island (YI) in the Japan MU, and Taiwan, Hong Kong, Lanyu Island, and Xisha Island (XI) in the IP MU (Fig. 1). According

to *Okuyama et al. (2009)*, the dispersion of turtles that hatch in these rookeries occurs through passive transport facilitated by the Kuroshio Current, Kuroshio branches, and other components of the northwestern Pacific Gyre. The green turtles observed at the feeding grounds located in the East China Sea, specifically the YI and CRI, originated from the YI, Southeast Asia, Micronesia, and Marshall Island rookeries in the Western Pacific region (*Nishizawa et al., 2013*). In contrast, turtles found at feeding grounds located in the northwestern region of mainland Japan, such as Nomaike, Muroto, Kanto, and Sanriku, originated mainly from the OGA, although some originated from the CRI and YI (*Hamabata et al., 2015*). The Sanriku feeding ground in the northernmost region also reported the presence of some Hawaiian turtles (*Nishizawa et al., 2014*). Within the northwestern Pacific region, there are additional important feeding grounds for green turtles, namely the Northeast China Sea, which spans Japan, China, and Korea, including the West (Yellow), South, East (Japan), and Jeju Seas. Nevertheless, the current understanding of the genetic composition and source rookeries of turtle populations in this region remains elusive.

*Chelonia mydas* is the most frequently observed species in Korean waters (*Kim et al., 2017*). Juvenile and adult green turtles are observed at an average rate of 5–10 turtles per year. Ecological and satellite tracking studies have been conducted to gain insights into the physical attributes, habitat preferences, and movement and migration patterns of green turtles (*Jang et al., 2018*; *Kim et al., 2022*, *2024*). Such studies have shown that green turtles actively use the sea around the Korean Peninsula and migrate to breeding grounds in both the Japan MU and the IP MU. Nevertheless, the current understanding of the genetic composition, genetic diversity, and source rookeries remains limited and requires immediate investigation. Most sea turtles inhabiting the marine ecosystems surrounding the Korean Peninsula are acquired through the processes of by-catch and stranding (*Kim et al., 2017*). Local feeding populations of these turtles are jeopardized by a variety of issues, including fishing, construction, and pollution (*Moon et al., 2009*; *Kim et al., 2017*). Understanding the source rookeries and MUs of green turtles, for which the Republic of Korea has implemented various government conservation efforts (*Kim et al., 2022*, *2024*; *Moon et al., 2022*), is crucial for the long-term conservation of sea turtles. The inclusion of relevant studies was crucial and required immediate attention.

To evaluate the genetic vulnerability of the Korean population, we verified genetic composition and genetic diversity, as well as mixed stock analysis (MSA) to infer the source rookery of green turtles caught as bycatch or found stranded in feeding grounds around the Korean Peninsula. Our results provide crucial insights into the rookeries and MUs that are the focus of conservation efforts for endangered green turtles in the northwestern Pacific Ocean.

## MATERIALS AND METHODS

### Sampling

Between August 5, 2014, and August 26, 2022, we sampled green turtles (*C. mydas*) that were reported to the National Marine Biological Resources Center (NMBRC) as being stranded in Korean territorial seas or caught unintentionally (*e.g.*, bycatch) during fishery
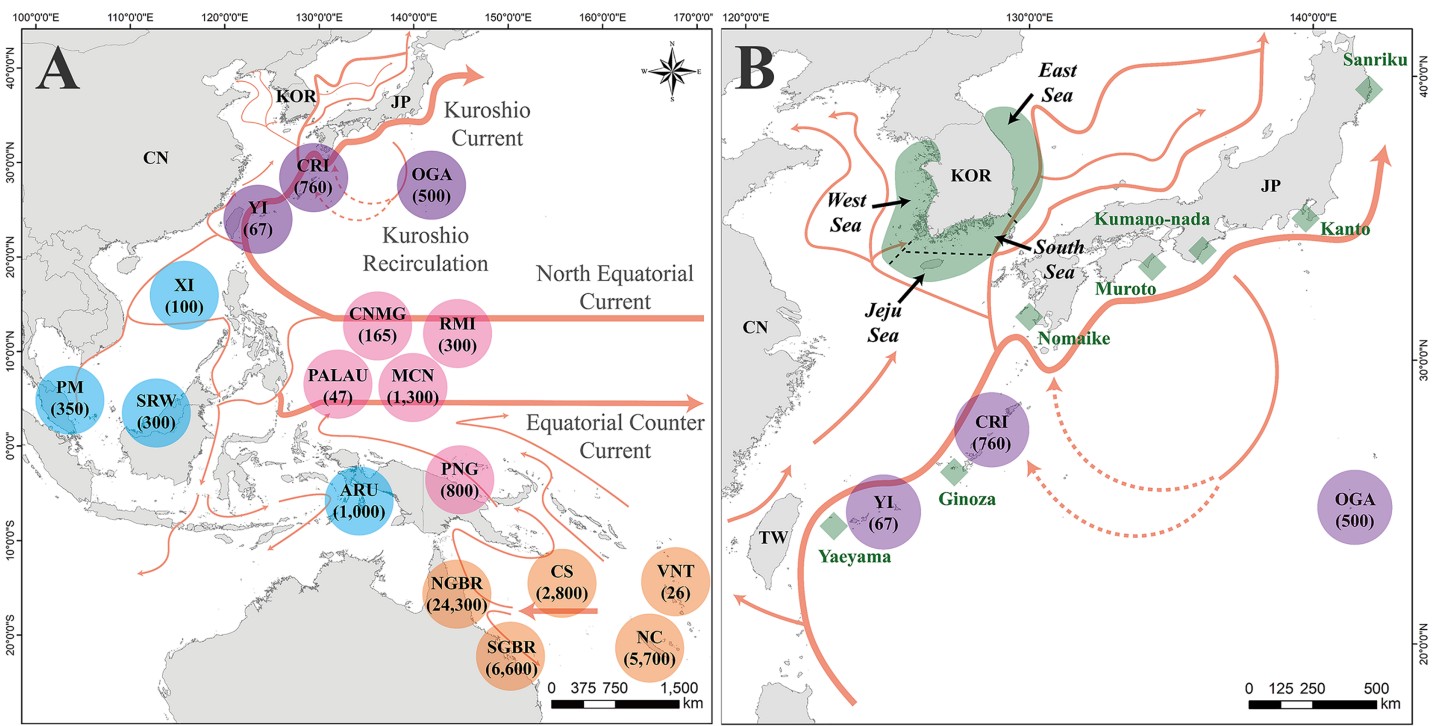

**Figure 1 Map of the study area, location of the rookeries, and feeding grounds within *Chelonia mydas* management units.** (A) Location of *Chelonia mydas* rookeries in Pacific regions for mixed stock analysis (MSA). The circle represents the location of *C. mydas* rookeries. The numbers under the rookery names indicate the population size of the rookery. Purple represents rookeries in the Japan management unit (MU), blue represents in the Indo-Pacific (IP) MU, pink represents in the Central Western Pacific (CWP) MU, and orange represents in the South Western Pacific (SWP) MU. The arrow on the map indicates the direction of the ocean current. (B) A map of the *C. mydas* feeding grounds in the Republic of Korea and Japan. Green area surrounding the Korean Peninsula indicates four feeding grounds, investigated in this study . The demarcation between the feeding grounds was established based on imaginary maritime boundaries, indicated by dotted black lines. Green rhombus indicates Japanese feeding grounds. The Kuroshio Current and its branch currents are also shown in (B). This map was generated based on GADM data (https://gadm.org/) using ArcGIS v. 10.1 (ESRI, Redlands, California, USA). Ocean currents direction in (A and B) reference to *Imawaki et al. (2001)*, *Misuguchi et al. (2007)* and *Hu & Wang (2016)*. NC, New Caledonia; VNT, Vanuatu; SGBR, Southern Great Barrier Reef; CS, Coral Sea; NGBR, Northern Great Barrier Reef; PNG, Papua New Guinea; ARU, Aru; MCN, Micronesia; RMI, Republic of the Marshall Islands; CNMG, CNMI/Guam; PALAU, Palau; SRW, Sarawak; PM, Peninsula Malaysia; XI, Xisha Island; YI, Yaeyama Island; CRI, Central Ryukyu Island; OGA, Ogasawara; TW, Taiwan (Republic of China); CN, China; KOR, Republic of Korea; JP, Japan.

work. For deceased turtles, we collected a small portion of the pectoral muscle during postmortem examination at NMBRC. For live turtles, 1 mL of blood was collected from the dorsal cervical vessels located in the lateral dorsal cervical region using a 23-gauge needle.

Blood samples were kept on ice, transported to NMBRC, and centrifuged to separate blood cells. After sampling, we released the live turtles to the ocean following appropriate recovery treatments such as trauma, malnutrition, exhaustion, and external parasites at institutions specializing in rescuing and treating marine animals. The final release decision was approved by the Marine Animal Protection Committee of the Republic of Korea based on the Marine Animal Release Evaluation Checklist (Legal notice #2020-198, Ministry of Oceans and Fisheries, Republic of Korea). The collected samples were kept either at −20 °C or at 4 °C in 99.5% ethanol until the DNA extraction. A voucher number was allocated to all samples stored in dry or refrigerated storage at the National Marine Biodiversity Institute of Korea (MABIK).

We acquired supplementary individual data during field sampling. The curved carapace length (CCL) was measured up to 0.1 cm using a tape measure (KMC-32; Komelon, Busan, Republic of Korea), and the body weight was measured up to 0.5 kg using a CAS precision scale (CPS PLUS; CAS, Seoul, Republic of Korea). The sex was determined based on the well-developed tail and front claws of males. If this was not possible, the sex was designated as unknown. Turtles were categorized into two maturity groups, juveniles and adults, based on their CCL measurement of 700 mm (*Green, 1993*). We also documented the location of sample acquisition, including GPS coordinates, if accessible, the year and month of capture, and the capture area (Table S1). However, the information on some turtles is incomplete, as certain details are missing due to the limitations of the 8-year study period.

The sample collection and all the experimental procedures were reviewed and approved by the Institutional Animal Care and Use Committee of the National Marine Biodiversity Institute of Korea (MABIK IACUC reference number MAB-23-02).

## DNA amplification and sequencing

Genomic DNA was extracted from the tissue and blood samples using the Qiagen DNeasy Blood and Tissue Kit according to the manufacturer's protocol (Qiagen, Hilden, Germany). For the genetic study, we amplified a partial sequence of the control region (CR, 860 bp) of mitochondrial DNA, a regularly used marker for studying genetic variation in green turtles (*Hamabata et al., 2015*; *Li et al., 2023*).

We conducted the polymerase chain reaction (PCR) using the primers LCM15382 (5′-GCTTAACCCTAAAGCATTGG-3′) and H950 (5′-GTCTCGGATTTAGGGGTTT-3,' *Abreu-Grobois et al., 2006*) to amplify the gene in a SimpliAMP Thermal Cycler (Applied Biosystems, California, USA). The PCR solution consisted of 10 μL of 2×TOPsimpleTM PreMIX-nTaq (Enzynomics, Incheon, Republic of Korea), 1 μL of template DNA, and 0.5 μL of each 10 pmol forward and reverse primers, and finally adjusted with molecular biology-grade water (HyClone, Logan, UT, USA) to the final volume of 20 μL. PCR products were confirmed on a 1% agarose gel and sequenced by Macrogen (Macrogen Inc., Seoul, Republic of Korea). We visually inspected and aligned the obtained sequences using MUSCLE (*Edgar, 2004*) and trimmed the sequences using Geneious Prime v.2022.0.2 (https://www.geneious.com). We finally used 796 bp sequences of the CR gene in the analyses.

## Genetic composition analysis

We determined the haplotypes of the 40 green turtles by conducting a nested BLAST within the NCBI in Geneious Prime v.2022.0.2. We assigned the names of haplotypes based on the Pacific haplotype (CmP) and Atlantic haplotype (CmA) nomenclature as specified by the Archie Carr Center for Sea Turtle Research (ACCSTR). We subsequently calculated the number of polymorphic sites, haplotype diversity ($h$), and nucleotide diversity ($\pi$) of the samples using DnaSP v.6 (*Rozas et al., 2017*).

To evaluate the genetic relationships between the sampled turtles and other turtles within the known 34 breeding (rookery) populations worldwide, we constructed a

median-joining haplotype network of the CR haplotypes using PopART v.1.7 (https://popart.maths.otago.ac.nz/; *Bandelt, Forster & Röhl, 1999*; *Leigh & Bryant, 2015*). Haplotype sequences and the number of turtles for each haplotype in the populations were obtained from GenBank and ACCSTR, as well as from published papers (Table S2). After aligning all the sequences used in the haplotype network, they were aligned to 754 bp, unlike the sequences aligned for 40 Korean green turtles (794 bp).

In this study, we assigned the clade name regionally (Table S2) based on previous studies (*Wallace et al., 2010*; *Jensen et al., 2019*) because the regional boundary of these genetic clades aligns well with the existing MU of green turtles in our study region. Three breeding populations were included in the Eastern Caribbean, North Western Atlantic, and South Atlantic clades, one in the Mediterranean clade (so far Atlantic), five in the Central and Eastern Pacific clade (CEP), two in the CSP clade, five each in the SWP and CWP clades, four in the IP clade, and three in the Japan clade (*Shamblin et al., 2012*; *Dutton et al., 2014*; *Hamabata, Kamezaki & Hikida, 2014*; *Read et al., 2015*; *Shamblin et al., 2015a*, *2015b*; *Joseph & Nishizawa, 2016*; *Hamabata et al., 2020*; *Barbanti et al., 2019*; *Dolfo et al., 2023*; *Li et al., 2023*).

## Mixed stock analysis

The mixed stock analysis (MSA) method uses Bayesian approaches to estimate the contribution of multiple source populations to the feeding ground for knowing their origin (*Bolker et al., 2007*). For the MSA, we utilized a many-to-many approach while simultaneously considering the various characteristics of the study population and the many source rookeries (*Bolker et al., 2007*; *Coșier & Petrescu-Mag, 2013*; *Hamabata et al., 2015*). Both a flat MSA method, which evenly weighs contributions from all rookeries, and a weighted MSA based on rookery size, were applied. Population size data for the seven rookeries examined (Table S3) were obtained from previous studies and available reports (*Dethmers et al., 2006*; *The State of the World's Sea Turtles (SWOT), 2011*; *Hamabata, Kamezaki & Hikida, 2014*; *Hamabata et al., 2020*). We did not conduct a weighted MSA on the rookery distance because the migration route of green turtles in this region remains unclear.

We categorized the available source rookeries in the Western Pacific Ocean into seven rookeries, considering the MUs and genetic clades of green turtles (*Wallace et al., 2010*; *Jensen et al., 2019*) and as well as our preliminary MSA results and haplotype network analyses. Of the 34 breeding (rookery) populations used in the haplotype network analysis, we included 17 rookeries within four regional MUs (Fig. 1, Table S3): the SWP, CWP, IP, and the Japan MU. Four rookeries (Xisha Island (XI) in the IP MU; Yaeyama Island (YI), Central Ryukyu Island (CRI), and Ogasawara (OGA) in the Japan MU) were used as independent units in our MSA analyses because they were geographically close to the study area. We expected to obtain a better resolution in the results of the source rookery investigation.

We performed MSA based on the combined group, sexual maturity group (juvenile and adult), seasonal group (spring, summer, fall, and winter), and specific feeding ground group where the turtles were collected (West (Yellow) Sea, South Sea, East Sea, and Jeju

Sea). Such detailed analyses can provide better information regarding where and when conservation efforts should be undertaken for specific target groups. The demarcation between the West and South Seas and between the South and East Seas was established based on imaginary maritime boundaries along Jindo Island and Ulsan City, respectively (Fig. 1). For seasonal analysis, we conducted MSA with only summer and fall data because we had only two samples in spring and one in winter. Because of the low temperatures, during winter and spring, observations of sea turtles are rare in the Republic of Korea (*Kim et al., 2017*). In addition, we removed the West Sea from the analysis of specific feeding grounds because only one sample was available.

All MSA models were created with 100,000 iterations of Markov chain Monte Carlo (MCMC) using the mixstock package in R version 4.3.1, and the first 50,000 runs were removed as burn-in (*Bolker et al., 2007*; *R Core Team, 2016*). All MSA analyses were conducted exclusively when the convergence value in the Gelman and Rubin Shrink Factor was less than 1.2, indicating that the data reached a stable state (*Pella & Masuda, 2001*).

## RESULTS

We collected samples from 40 turtles, comprising 21 adults, 15 juveniles, and 4 of unknown maturity. Among these, 24 were female, 3 were male, and the sex of the 13 turtles remained undetermined. Of the 40.9% that were sampled between June and October, the highest number we collected in August (25%). Seasonally, 23 turtles were sampled in summer (June–August), 14 in fall (September–November), 2 in spring (March–May), and 1 in winter (December–February) (Table 1). Geographically, the East Sea accounted for 14 turtles, followed by 13 in the Jeju Sea, 12 in the South Sea, and 1 in the West (Yellow) Sea (Tables 1 and S1).

### Genetic composition analysis

Ten distinct CR haplotypes (GenBank accession numbers PP691472–PP691481) were identified from 41 polymorphic sites (15 singleton variable sites and 26 parsimony-informative sites) (796 bp, Table 1). All haplotypes were concordant with previously documented haplotypes in this species (*Hamabata, Kamezaki & Hikida, 2014*; *Hamabata et al., 2015*, *2020*). Orphan haplotypes were not observed. Haplotype Cmp39.1, found in 18 turtles (45.0%), exhibited the highest frequency, followed by haplotype Cmp50.1, detected in six turtles (15.0%). Each of the three turtle individuals (7.5%) had haplotypes Cmp49.1, Cmp121.1, or Cmp128.1. Haplotypes Cmp53.1 and Cmp127.1 were found in two turtles (5.0%), and Cmp54.1, Cmp70.1 and Cmp79.1 were found in one turtle (2.5%) (Table 1). The haplotype diversity was 0.771 ± 0.060, whereas the nucleotide diversity was 0.01338 ± 0.00190.

Within the haplotype network, six haplotypes (CmP39.1, CmP70.1, CmP79.1, CmP121.1, CmP127.1, and CmP128.1) were assigned to the Japan clade (Fig. 2). The other three haplotypes (CmP49.1, CmP50.1, and CmP53.1) were associated with the IP clade but were exclusively found in the Japanese population except the CmP49.1 haplotype that widespread in the IP MU. The last haplotype (CmP54.1) grouped with the CWP clade and was previously found only in the Japanese population (Fig. 2).

Table 1 Distribution of control region haplotypes (CR, 796 bp) of 40 green turtles (*Chelonia mydas*) based on the maturity, season, and specific feeding ground in the study area.

| Haplotype | Maturity | | | Season | | | | Feeding ground | | | | Total |
|---|---|---|---|---|---|---|---|---|---|---|---|---|
| | Juvenile | Adult | Unknown | Spring | Summer | Autumn | Winter | West sea | East sea | South sea | Jeju sea | |
| CmP39.1 | 6 | 10 | 2 | 1 | 10 | 7 | | | 4 | 9 | 5 | 18 |
| CmP49.1 | 1 | 1 | 1 | | 2 | 1 | | | 3 | | | 3 |
| CmP50.1 | 3 | 3 | | 1 | 2 | 3 | | | 2 | 2 | 2 | 6 |
| CmP53.1 | 2 | | | | 1 | 1 | | 1 | | | 1 | 2 |
| CmP54.1 | | 1 | | | 1 | | | | 1 | | | 1 |
| CmP70.1 | | 1 | | | 1 | | | | 1 | | | 1 |
| CmP79.1 | 1 | | | | 1 | | | | | | 1 | 1 |
| CmP121.1 | | 2 | 1 | | 1 | 2 | | | 1 | 1 | 1 | 3 |
| CmP127.1 | | 2 | | | 1 | | 1 | | 1 | | 1 | 2 |
| CmP128.1 | 2 | 1 | | | 3 | | | | 1 | | 2 | 3 |
| Total | 15 | 21 | 4 | 2 | 23 | 14 | 1 | 1 | 14 | 12 | 13 | 40 |

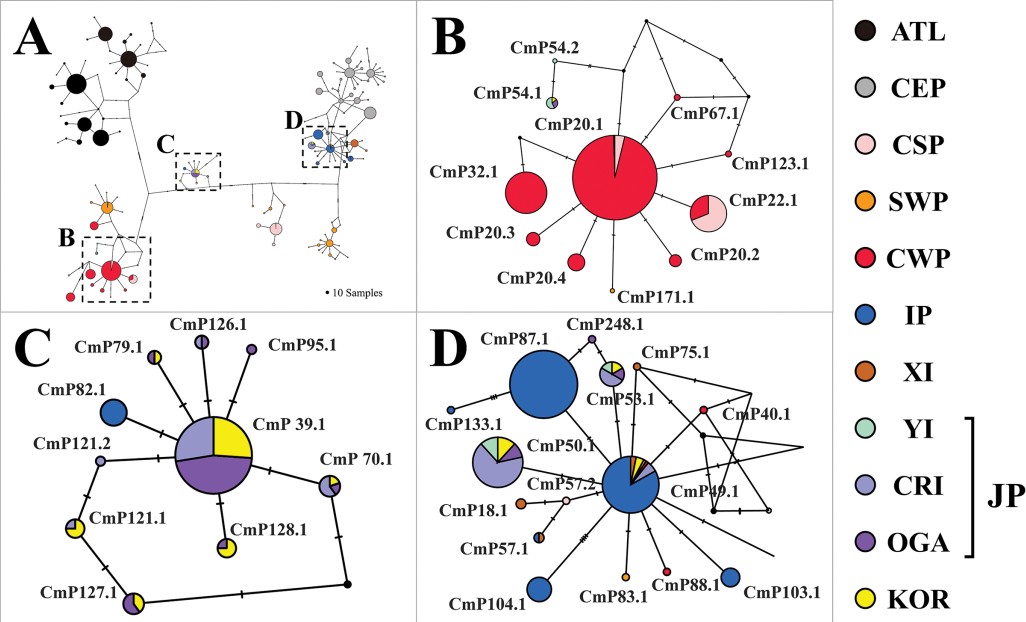

Figure 2 Haplotype network of the mitochondrial control region (CR) sequences (754 bp) of *Chelonia mydas* among 34 rookery populations worldwide. A number of mutations between haplotypes are illustrated by dashes in connecting lines. The size of the circle means the sample size of each haplotype. We presented regional clade names on the network, considering known management units of green turtles (*Jensen et al., 2019*; *Wallace et al., 2010*). Yellow (KOR) indicates the haplotypes identified in this study. The ATL represents the haplotypes found in the North Western Atlantic, South Atlantic, Eastern Caribbean, and Mediterranean. CEP, Central and Eastern Pacific; CSP, Central South Pacific; SWP, South Western Pacific; CWP, Central Western Pacific; IP, Indo-Pacific; XI, Xisha Island; YI, Yaeyama Island; CRI, Central Ryukyu Island; OGA, Ogasawara; JP, Japan; KOR, Republic of Korea.

## Mixed stock analysis

In the flat MSA results, rookeries (YI, CRI, and OGA) in the Japan MU accounted for 87.7% of the 40 turtles examined (Fig. 3A, Table S4). Rookeries (IP and XI) in the IP MU contributed 7.4% of the total. A closer look at the Japan MU, OGA, and CRI rookeries contributed 41.9% and 37.8% of the 40 turtles examined, respectively. When considering rookery size, the contribution ratio changed slightly to 46.1% for CRI and 38.7% for OGA.

In the flat MSA results of the season, OGA and CRI contributed 48.0% and 17.5% of the 23 turtles in the summer, respectively, whereas CRI, OGA, and YI contributed 38.6%, 14.9%, and 14.7% of the 14 turtles in the fall, respectively (Fig. 3B, Table S4). When considering rookery size, the contribution of OGA increased to 51.0% in summer, whereas CRI increased to 57.2% in autumn. In the comparison of specific feeding grounds, OGA mainly explained the turtles in the Jeju Sea (35.0%), whereas CRI did so in the South Sea (32.5%) and East Sea (20.8%) (Fig. 3C, Table S4). Specifically, 20.4% of the 14 turtles in the East Sea came from the Indo-Pacific rookery, and 17.8% came from the YI rookery. When considering rookery size, the contribution of OGA increased to 40.6% in the Jeju Sea and 22.6% in the East Sea. The contribution of the CRI increased in all feeding grounds. In addition, we conducted MSA based on the maturity groups of the juvenile and adult. The contribution patterns were similar between juveniles and adults and overall to the combined data (Fig. 3D, Table S4). In the flat MSA results, OGA and CRI contributed 29.8% and 19.8%, respectively, in juveniles and 33.9% and 36.2%, respectively, in adults. When considering rookery size, the contribution of CRI increased to 36.5% in juveniles and 44.5% in adults (Fig. 3D, Table S4).

## DISCUSSION

Most green turtles in the feeding grounds around the Korean Peninsula have a genetic composition belonging to the Japan clade, although some have sources from the IP and CWP clades. This result is consistent with that of a previous phylogenetic study (*Jensen et al., 2019*). Haplotype network analysis revealed that six haplotypes were associated with the Japan clade, whereas the other four haplotypes were grouped with either the IP or CWP clade. However, three of these four haplotypes have been identified previously only in the Japanese population. Only one haplotype (Cmp49.1) was found across the populations in Japan and the IP MU. Overall, 9 of the 10 haplotypes (37 of 40 turtles) were found in the Japanese MU. However, two pieces of evidence suggest that some turtles also originate from either the IP or CWP MU. First, as previously described, one haplotype (3 of 40 turtles, Cmp49.1) was found in populations across Japan, IP, CWP, and SWP MUs, and was the main haplotype in the IP clade (*Jensen et al., 2019*). Second, previous satellite tracking studies have shown that green turtles in Korean waters have migrated to Hainan Island in the Indo-Pacific Ocean (*Kim et al., 2022*, *2024*). The genetic diversity of green turtles (0.77) in the study area was similar to the reported genetic diversity (0.65 – 0.88) in seven Japanese feeding grounds (*Nishizawa et al., 2013*, *2014*; *Hamabata et al., 2015*), showing that protecting the feeding population in the Republic of Korea could be meaningful to conserve genetic diversity of green turtles in northwestern Pacific Ocean.

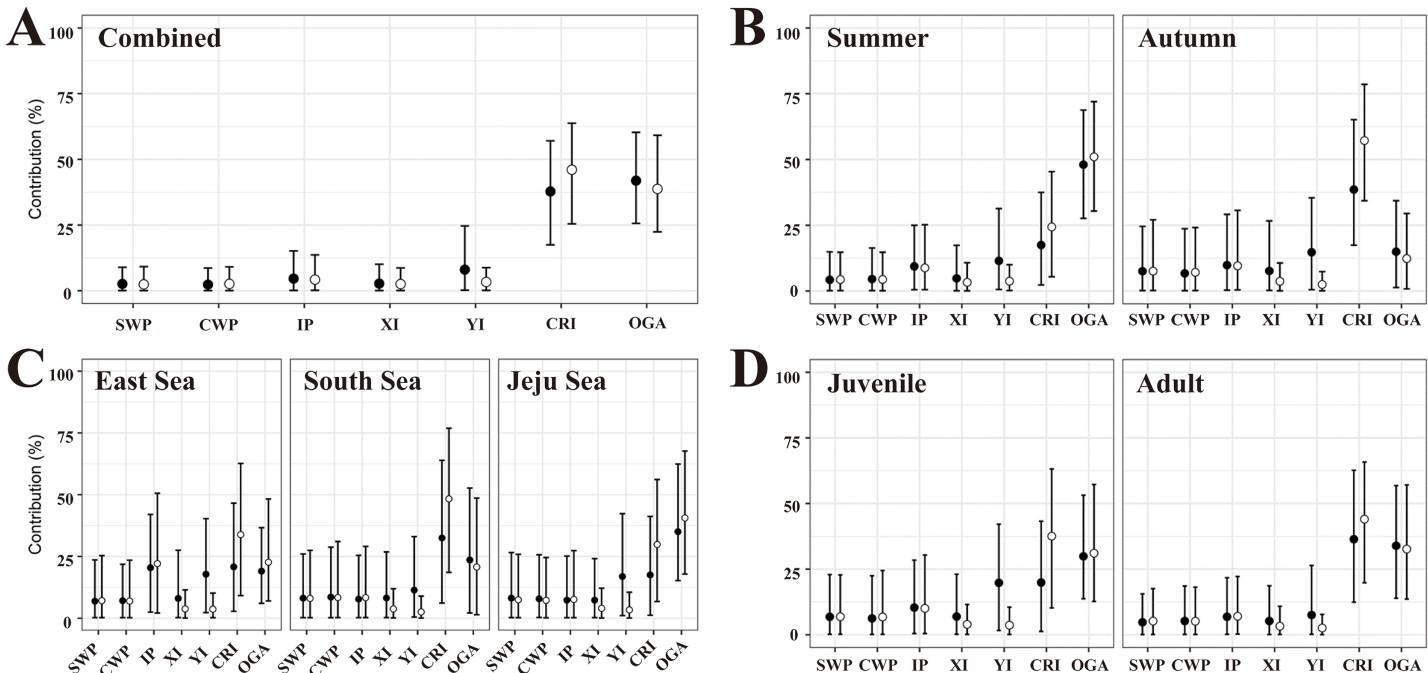

**Figure 3 The contributions of seven rookeries on the stock of the feeding grounds were studied based on the combined data (A), season (B), specific feeding ground (C), and maturity (D), in the mixed-stock analysis (MSA).** We categorized the rookery unit based on known genetic clades (*Jensen et al., 2019*) and the management unit (*Wallace et al., 2010*) of green turtles (*Chelonia mydas*) and included 17 individual breeding populations (Table S3). Points are mean estimates, and whiskers indicate 2.5% and 97.5% credibility intervals. Flat MSA results are indicated black and weighted (rookery size) MSA results are indicated white. SWP, South Western Pacific; CWP, Central Western Pacific; IP, Indo-Pacific; XI, Xisha Island; YI, Yaeyama Island; CRI, Central Ryukyu Island; OGA, Ogasawara.

Green turtles in the feeding grounds in the vicinity of the Korean Peninsula originated from two geographically distant rookeries: the OGA, situated in the northern area of the North Western Pacific gyre, and the CRI, located in the Central Western Pacific gyre. The OGA rookery made the largest contribution to the feeding ground, accounting for 41.9% of the total and the CRI had the second-largest contribution at 37.8%. Both the use of oceanic currents for migration and active movements between rookery and feeding grounds could explain such results. It is well known that ocean currents are major factors in sea turtle migration (*Okuyama et al., 2009*; *Nishizawa et al., 2013*). The OGA is well-linked to the study area through the Kuroshio recirculation in the northwestern Pacific (*Hurlburt et al., 1996*). The main Kuroshio Current and its branch currents offer good transportation options for the CRI (*Park et al., 2013*; *Zhong et al., 2021*). On the other hand, green turtles also actively move between their rookery and feeding grounds. Most turtles in feeding grounds along the southeastern coast of mainland Japan originated from close OGA rookery while turtles in the southern Ryukyu feeding grounds come from geographically close southern Ryukyu or the Indo-Pacific regions (*Nishizawa et al., 2013*; *Hamabata et al., 2015*).

The contribution of rookeries to feeding grounds was influenced by the season and specific feeding grounds in the study area. First, there was a clear seasonal pattern in the number of turtles collected and the influx from different rookeries. The majority of the

turtles were collected between June and November. Conversely, only three turtles were collected between December and May, indicating that the majority of turtles came from summer and fall, likely because of low temperatures during winter (*Kim et al., 2017*). It is well known that the feeding activity of sea turtles largely depends on water temperature (*Reisser et al., 2013*). In summer, most turtles originated from the OGA. The Kuroshio recirculation, generated by the Kuroshio Current in the OGA region, may facilitate the movement of green turtles toward the west or north direction (*Hurlburt et al., 1996*). This allows them to reach the Kuroshio Current, which flows into the northern parts of the Ryukyu Islands or the southeastern sea of mainland Japan. The turtles then arrive at feeding grounds in the Republic of Korea. In particular, given that the dispersion of post-breeding green turtles at OGA occurs in summer (*Kondo et al., 2017*), there is a potential for augmentation in the westward and northward migrations of these turtles.

In contrast, the CRI rookery accounted for the largest proportion of turtles in the fall. When considering rookery size, it increased by more than 50%. There were two potential variables for this pattern. First, during this time of year, the Kuroshio Current, which aligns with the northward surface wind direction and monsoon activity in the area, increases in speed and volume (*Isobe, 1999*; *Zhong et al., 2021*). Moreover, the current ran closer to the shores of the Ryukyu Islands. Such changes could increase the chance of green turtles moving northward owing to currents in the East China Sea. The Kuroshio Current is widely recognized for its substantial influence on the migration patterns of diverse marine organisms in the Western Pacific (*Andres et al., 2015*). Second, typhoons are often generated in July and August (*Choi, Cha & Kim, 2012*). Typhoons originating in the southern Pacific Ocean travel northerly and pass across the southern and central Ryukyu Islands (*Choi, Cha & Kim, 2012*). Typhoons could potentially enhance the chances of green turtles migrating to the feeding grounds in the study area. Previous studies have shown that typhoons transport diverse flora and fauna from the East China Sea to the Korean Peninsula (*Osozawa et al., 2021*; *Lee et al., 2023*). Our findings suggest that the contributions of the OGA and CRI rookeries to the study area have distinct seasonal patterns, largely based on the activity of the Kuroshio Current and its branches.

The influx of green turtles into the study area fluctuated depending on the specific feeding grounds within the study area. The OGA rookery primarily contributed to the Jeju Sea regardless of rookery size, whereas the CRI rookery explained more turtles in the South and East Seas than the OGA. In particular, the YI and rookeries in the IP MU contributed some of the turtles (46.2%) in the East Sea in the flat model. The observed pattern could also be attributed to regional water influx by ocean currents, such as the Kuroshio Current and its branch currents, including the Tsushima Current, as well as the Kuroshio recirculation near the OGA. The Kuroshio branch currents, such as the Tsushima Current, receive water from the Kuroshio Current and, in autumn, transport the water more directly to the East Sea (*Isobe, 1999*). This flow of water may bring turtles to the South Sea and East Sea from rookeries such as the YI and CRI on the Ryukyu Islands and/or the IP MU. From the OGA to the Jeju and South Seas, the southeastern coastal migration route of mainland Japan has the potential to transport turtles to the feeding grounds studied. The Kuroshio recirculation may first transport OGA turtles to the southeastern shore of

mainland Japan (*Hurlburt et al., 1996*) and then follow the coastline westward to reach the Jeju Sea. On the other hand, in our previous satellite tracking study, green turtles released in the Jeju Sea in the Republic of Korea used across close feeding grounds including the Jeju Sea, the East Sea, and the South Sea in the Republic of Korea as well as sea nearby Kyushu Island in Japan (*Kim et al., 2024*). These areas are abundant in seaweed, which is food for green sea turtles (*Kim et al., 2021*). Also, as mentioned above, *Hamabata et al. (2015)* and *Nishizawa et al. (2013)* showed active migration of green turtles between close rookery and feeding grounds. These active movements could also affect the overall appearance of green turtles, in particular, in specific feeding grounds around the Korean Peninsula.

## CONCLUSIONS

In summary, the feeding grounds in the study area were utilized by both juvenile and adult green turtles, originating primarily from the Japan MU and partially from the IP and CWP MU in the northwestern Pacific Ocean. The turtle population in the study area consisted of a distinct combination of two main geographically distant rookeries, the OGA and the CRI. The influx from these two rookeries differed based on the season and specific feeding grounds in the area. In addition, we suggest that the Kuroshio Current and its branches are crucial for the migration of green turtles to the northwest Pacific Ocean. The results of our study provide crucial insights into rookeries and MUs, which are the focus of conservation efforts in the Republic of Korea. They also support the collaboration between local governments and national parties in demographic information exchange and recovery projects to conserve green turtles effectively.

## ACKNOWLEDGEMENTS

Thanks to Aqua Planet Yeosu, Aqua Planet Jeju, Sealife, Korea National Maritime Museum, and National Institute of Ecology for the sampling help.

### Funding

This study was supported by grants from the National Marine Biodiversity Institute of Korea (2024M00300 and 2024E00300) funded by the Ministry of Oceans. The funders had no role in study design, data collection and analysis, decision to publish, or preparation of the manuscript.

### Grant Disclosures

The following grant information was disclosed by the authors:
National Marine Biodiversity Institute of Korea: 2024M00300 and 2024E00300.

### Competing Interests

The authors declare that they have no competing interests.

## Author Contributions

- Min-Woo Park conceived and designed the experiments, performed the experiments, analyzed the data, prepared figures and/or tables, authored or reviewed drafts of the article, and approved the final draft.
- Il-Hun Kim conceived and designed the experiments, analyzed the data, prepared figures and/or tables, authored or reviewed drafts of the article, and approved the final draft.
- Jaejin Park performed the experiments, analyzed the data, prepared figures and/or tables, authored or reviewed drafts of the article, and approved the final draft.
- Changho Yi conceived and designed the experiments, analyzed the data, authored or reviewed drafts of the article, and approved the final draft.
- Min-Seop Kim conceived and designed the experiments, performed the experiments, analyzed the data, authored or reviewed drafts of the article, and approved the final draft.
- In-Young Cho conceived and designed the experiments, performed the experiments, analyzed the data, authored or reviewed drafts of the article, and approved the final draft.
- Il-Kook Park analyzed the data, prepared figures and/or tables, authored or reviewed drafts of the article, and approved the final draft.
- Hee-Jin Noh conceived and designed the experiments, analyzed the data, authored or reviewed drafts of the article, and approved the final draft.
- Sang Hee Hong conceived and designed the experiments, analyzed the data, authored or reviewed drafts of the article, and approved the final draft.
- Daesik Park conceived and designed the experiments, prepared figures and/or tables, authored or reviewed drafts of the article, and approved the final draft.

## Animal Ethics

The following information was supplied relating to ethical approvals (*i.e.*, approving body and any reference numbers):

The sample collection and all the experimental procedures were approved by the Institutional Animal Care and Use Committee of the National Marine Biodiversity Institute of Korea (MABIK IACUC reference number MAB-23-02).

## DNA Deposition

The following information was supplied regarding the deposition of DNA sequences:

The 10 CR haplotype sequences are available at GenBank: PP691472 to PP691481.

## Data Availability

The raw data are available in the Supplemental Tables.

## Supplemental Information

Supplemental information for this article can be found online at http://dx.doi.org/10.7717/peerj.17560#supplemental-information.

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
