# Peer review of "Identifying the source rookery of green turtles (Chelonia mydas) found in feeding grounds around the Korean Peninsula"

_PeerJ, doi:10.7717/peerj.17560_

## Round 0.1 · original submission · Major Revisions

Dear Dr. Park

The reviewers have commented on your manuscript. Based on the comments and suggestions of the expert reviewers, a major revision is needed for your article.

I would like to request that you check and correct the manuscript based on the reports.

Sincerely yours

·

Basic reporting

The article meets PeerJ standards. One issue that needs clarification is the raw data uploaded to GenBank (see below).

The English is clear, unambiguous, and professional. I have made some edits for grammar and clarity and will upload the pdf.

The references are sufficient to lay out the background and context

The article structure, figures, and tables are professional. There are DNA sequences produced in this paper, and it is unclear whether they were uploaded to a public database (i.e., GenBank). This should be done and the accession numbers included in the paper.

The paper is self-contained with relevant results to the hypothesis.

Experimental design

no comment

Validity of the findings

The paper is relatively straightforward, but an important study to understand the movement patterns of Chelonia mydas in Korea. The analyses are robust and statistically sound.

Actually, if the authors want, they could make stronger statements about the implications of their findings. Korea is an important location for the feeding of different rookeries and should be well protected. Also further emphasize the need for collaboration between different regions.

Reviewer 2 ·

Basic reporting

no comment

Experimental design

no comment

Validity of the findings

Ocean current could greatly impact the migration of sea turtles from their rookey to initial foraging ground. But for juvenile and adult turtle, they could also migrate seasonly between different foraing grounds, may not dirven by currents only. In your discussion, it seems that you only focus on migration between the birthplace and feeding grounds, without considering migration between different foraging grounds.

Additional comments

I have add my detail comments into the text.

Annotated reviews are not available for download in order to protect the identity of reviewers who chose to remain anonymous.

·

Basic reporting

The results are important and well written. However, there are small modifications that I suggest should be done to improve the comprehension of the document. The text accompanying Tables and Figures needs to be more explicit in terms of the meaning of acronyms and letters used. This is especially important for Figure 2, where it is very dififult to understand the contribution of present results to the regional haplotype network. In the case of Table 1, it should be redesign to asure that the totals are the same, maybe a third columns with the missing data or, at least, explain te reason of the difference in a note. In the Supplemental Table S2 I suggest that the references should be numbered, and those numbers placed in the corresponding information in the table, in order to better establish the nexus between the genetic information and their source.

Experimental design

No comment

Validity of the findings

No comment

---

## Round 0.2 · accepted · Accept

Dear Dr. Park,

I would like to thank you and your co-authors for making the corrections and changes requested by the reviewers. I read and checked carefully your valuable article and I am happy to inform you that your article has been accepted for publication in PeerJ.

Best regards

Reviewer 2 ·

Basic reporting

No comment

Experimental design

No comment

Validity of the findings

No comment

Additional comments

I'm satisfied with the careful revisions of authors.

·

Basic reporting

The manuscript is, in my opinion very well written, and it was improved with all the changes made after the revision. Figures are better understood after adding the information of localities

Experimental design

The experimental design is adequate

Validity of the findings

Findings presented in this manuscript are very important and supported by an extensive data gathering and discussion of the results